# Action Observation Therapy for Arm Recovery after Stroke: A Preliminary Investigation on a Novel Protocol with EEG Monitoring

**DOI:** 10.3390/jcm12041327

**Published:** 2023-02-07

**Authors:** Sara Boni, Martina Galluccio, Andrea Baroni, Carlotta Martinuzzi, Giada Milani, Marco Emanuele, Sofia Straudi, Luciano Fadiga, Thierry Pozzo

**Affiliations:** 1School of Medicine, Ferrara University, 44121 Ferrara, Italy; 2Iit@Unife Center for Translational Neurophysiology, Istituto Italiano Di Tecnologia, 44121 Ferrara, Italy; 3Doctoral Program in Translational Neurosciences and Neurotechnologies, Ferrara University, 44121 Ferrara, Italy; 4Neuroscience and Rehabilitation Department, Ferrara University Hospital, 44121 Ferrara, Italy; 5Neuroscience and Rehabilitation Department, Ferrara University, 44121 Ferrara, Italy

**Keywords:** stroke, rehabilitation, action observation, EEG

## Abstract

This preliminary study introduces a novel action observation therapy (AOT) protocol associated with electroencephalographic (EEG) monitoring to be used in the future as a rehabilitation strategy for the upper limb in patients with subacute stroke. To provide initial evidence on the usefulness of this method, we compared the outcome of 11 patients who received daily AOT for three weeks with that of patients who undertook two other approaches recently investigated by our group, namely intensive conventional therapy (ICT), and robot-assisted therapy combined with functional electrical stimulation (RAT-FES). The three rehabilitative interventions showed similar arm motor recovery as indexed by Fugl-Meyer’s assessment of the upper extremity (FMA_UE) and box and block test (BBT). The improvement in the FMA_UE was yet more favourable in patients with mild/moderate motor impairments who received AOT, in contrast with patients carrying similar disabilities who received the other two treatments. This suggests that AOT might be more effective in this subgroup of patients, perhaps because the integrity of their mirror neurons system (MNS) was more preserved, as indexed by EEG recording from central electrodes during action observation. In conclusion, AOT may reveal an effective rehabilitative tool in patients with subacute stroke; the EEG evaluation of MNS integrity may help to select patients who could maximally benefit from this intervention.

## 1. Introduction

Motor impairment of the upper limb is one of the most common long-term disabilities after a stroke [1]. Over the years, many rehabilitation strategies have been applied to maximise the functional outcome in stroke survivors, including conventional physical therapy, constraint-induced movement therapy [2], and robot-assisted therapy (RAT) [3], among the most successful. Most rehabilitative programs engage motor-related areas through repetitive supervised training, which has improved upper limb motor recovery [4,5,6]. However, interventions such as movement therapy or robot-assisted therapy are only viable when paretic patients with subacute stroke regain active movements of the impaired limb. Moreover, fatigue quickly limits the application of rehabilitative interventions requiring active patient participation.

To address these limitations, this preliminary investigation aims at comparing arm-intensive conventional therapy and RAT with action observation therapy (AOT). The rationale behind this rehabilitative intervention relies on the well-established evidence that motor areas are recruited while performing actions and observing them [7]. Evidence of motor activation during action observation (AO) was originally gathered in primate studies, whereby a population of neurons in the premotor area F5—henceforth known as mirror neurons—showed similar discharge when monkeys performed a goal-directed action and when they observed similar actions performed by other individuals [8,9]. Subsequently, neural populations with similar response properties were also discovered in humans using invasive and non-invasive techniques, such as electroencephalography (EEG), functional magnetic resonance imaging, positron emitted tomography, and transcranial magnetic stimulation [7,10,11,12,13,14]. This network of brain regions with mirror activation properties forms the Mirror Neuron System (MNS) [15].

Moving from this background, AOT protocols are designed to exploit the response properties of the residual MNS in the damaged brain to promote and accelerate motor recovery [16,17]. During an AOT intervention, patients are presented daily with motor tasks performed by healthy subjects. Repetitive observation of others’ actions engages patients’ MNS, thus prompting the rehearsal of lost or impaired motor patterns without actual movements being performed. Just as mirror neurons are recruited by both motor execution and observation, intensive and systematic (visual) exposure to actions performed by others is thought to establish a similar form of motor plasticity [18] as that induced by rehabilitative interventions involving active movements. Encouraging results have recently shown that AOT is effective in promoting motor recovery in stroke survivors [19,20], especially for the rehabilitation of the upper limb, possibly leading to a more favourable clinical outcome when compared to traditional rehabilitation [21,22,23].

In this work, we aimed to gather preliminary data comparing an AOT rehabilitative intervention in addition to conventional therapy, with two other approaches recently investigated by our group [24], namely intensive conventional therapy (ICT) and RAT combined with functional electric stimulation (RAT-FES). A previous clinical trial failed to demonstrate any differences between these two approaches in improving upper limb motor function [24]. Differently from these interventions, the AOT protocol employed in the current work carries the major advantage of requiring only a minimal active motor task to be performed by the patients, that is, to imitate a few of the presented actions when (randomly) asked by the experimenter. This was carried out to ensure that patients properly attended to the presented actions. Patients were otherwise requested to watch the presented actions while resting on a chair/wheelchair, thus minimizing the physical fatigue induced by the protocol.

Patients who received AOT also underwent EEG recordings to monitor the activation of motor-related cortical resources. Action observation elicits a well-established pattern of EEG activity in central electrodes known as mu (µ) (8–13 Hz) suppression, which is commonly used to index the activation of the MNS [25]. By examining µ suppression, we could assess the functional integrity of MNS [26] in each patient and relate this information with the clinical outcome of AOT.

## 2. Materials and Methods

### 2.1. Participants

Thirty-three patients (13 females; mean age = 60.97, SD = 12.87) diagnosed with unilateral stroke occurred within six weeks (i.e., 42 days, subacute stroke) (mean = 24.45, SD = 9.36), and scoring below 55 in the Fugl-Meyer Assessment of the Upper Extremity (FMA_UE), participated in the study. All patients were recruited at the Unit of Rehabilitative Medicine, Ferrara University Hospital. The stroke diagnosis was corroborated by neuroimaging. Patients with cerebellar stroke, severe visual, cognitive, or neuropsychological impairment, as well as hepatic, renal, cardiac, pulmonary diseases, pain visual analogue scale score over 7, pregnancy, or other severe conditions were excluded. Twenty-two out of the thirty-three patients recruited for the present study also participated in a previous study that compared rehabilitative interventions based on RAT-FES (10 patients; 5 females, mean age = 61.3, SD = 12.62) and ICT (12 patients; 5 females, mean age = 62.92, SD = 15.39) [24]. The remaining, newly recruited 11 patients underwent AOT (11 patients; 3 females, mean age = 58.55, SD = 10.72). The three groups matched for age, sex, and days from stroke to rehabilitation onset (AOT: mean = 27, SD = 9.32; ICT: mean = 23.67, SD = 10.80; RAT-FES: mean = 22.60, SD = 7.73). The demographic and clinical characteristics of the three treatment groups are shown in Table 1. All the procedures were conducted according to the ethical standard of the Declaration of Helsinki and approved by the local Ethics Committee [27]. Patients provided their written informed consent before undergoing the experimental procedures. The AOT trial protocol and the RAT-FES vs. ICT trial protocol were registered on ClinicalTrials.gov (NCT04622189 and NCT02267798, respectively).

### 2.2. Rehabilitative Intervention

All three rehabilitative strategies were carried out in daily sessions for three weeks, five days per week (Monday through Friday). Daily sessions lasted 100 min for each rehabilitative intervention. At the beginning (T0) and the end (T1) of the three-week rehabilitative interventions, patients’ motor function was evaluated by an experienced clinician by means of the FMA_UE and the Box and Block Test (BBT) (Figure 1).

Patients in the AOT group were comfortably seated on a chair/wheelchair with both arms lying relaxed on a table and were required to carefully observe videos displayed on a 24-inch monitor placed 65 cm in front of them. Videos randomly showed intransitive, non-object-related actions (e.g., meaningless finger movements) and transitive, objects-related actions (e.g., grasping a cup, lifting a plastic bottle, cutting an orange) from a first-person perspective, the one showing the strongest suppression during EEG recordings [29] (Figure 2). The actor in the video wore black sleeves and grey gloves. Videos lasted 4–16 s and were preceded by a 2-s countdown (3…2…1). The action began after 1 s of a static image depicting the actor’s upper limbs lying on a table. Every daily AOT session lasted about 40 min and included 120 videos divided into three blocks of 40 videos each. Each 40-min AOT session was followed by 60 min of conventional therapy (CT). Ten different videos were shown four times during each block. To ensure that patients attended the videos, they were also randomly required to imitate one of the observed actions four times after the 4th, 16th, 23rd, and 38th video of each block and asked to answer two questions displayed on the monitor after the 8th and 29th video (e.g., “was the last action already presented before in this block?”). The action that patients were required to imitate remained the same within each block. Visual actions changed from block to block, becoming more and more complex over the three weeks of treatment (e.g., tapping on the table in the first week, grasping an object in the second week, cutting an orange in the third week). Although the actor in the video was a right-handed experimenter, the videos were flipped according to each patient’s side affected by hemiplegia/hemiparesis. That is, in videos depicting uni-manual actions (e.g., grasping and lifting a cup), the actor always employed her dominant (right) hand; patients with right hemiplegia/hemiparesis were shown the original videos, while patients with left hemiplegia/hemiparesis were presented with a flipped version of the same videos as if actions were performed by a left-handed actor. Similarly, videos depicting bi-manual actions (e.g., unscrewing a plastic bottle) were all recorded by the same right-handed experimenter, being flipped according to the laterality of the motor impairment in each patient. This approach, while allowing a good standardisation of the stimuli presented to each patient, allowed us to maximally target the affected side in each patient.

The ICT and RAT-FES protocols are described in a previous work carried out by our group [24]. In brief, the patients in the ICT group received 100 min of specific exercises, including active, passive, and sensory exercises, as well as functional tasks. The RAT-FES group received 40 min of electric stimulation of the impaired hand, delivered by five electrodes placed on the extensor digitorum communis, extensor pollicis brevis, flexor pollicis longus, flexor digitorum superficialis, and thenar muscles. The stimulation intensity was set to provide comfortable and consistent activation of the extensor and flexor muscles normally used to achieve whole hand opening and functional grasping. FES was followed by 60 min of RAT performed using an end-effector device (Reo Therapy System, Motorika Medical Ltd., Caesarea, Israel) and focused on repetitive multidirectional reaching actions. All patients in the three treatment groups also received traditional multidisciplinary rehabilitation based on their personal needs and clinical characteristics.

### 2.3. EEG Recording and Pre-Processing

Patients who received AOT underwent EEG recordings during video presentation to monitor brain activity throughout each session of rehabilitative intervention. One patient did not agree to wear skin electrodes, so EEG data were collected from 10 out of 11 patients. EEG was recorded using a wireless Enobio EEG System (Neuroelectrics) with the left earlobe as a reference. The sampling frequency was 500 Hz. Electrical activity of the brain was recorded from two (C3 and C4) sites of the International 10–20 system. Electrode impedance was minimized by skin scrubbing and conductive gel.

EEG data were analysed using MATLAB (The MathWorks, Inc., Natick, MA, USA) and FieldTrip toolbox [30]. Data were first segmented offline in single-trial epochs lasting 8 s, from −3 s to +5 s, with respect to the onset of each video clip (i.e., static image of the hands). Trials showing muscular or ocular artefacts were rejected after visual inspection. For each epoch, time-frequency analysis was performed between 5 and 30 Hz, with a resolution of 0.125 Hz. A 400 ms time-interval within the period of static image (from 0.05 s to 0.45 s relative to the onset of static image presentation) served as a baseline, calculated as the logarithm (base ten) of each time-frequency point value divided by the average spectral power of the baseline at the same frequency.

As in previous works, the desynchronisation of the µ rhythm in central electrodes (C3 and C4) during action observation was analysed within three frequency bands: Low-alpha (8–10 Hz), high-alpha (10–13 Hz), and beta (13–30 Hz) [31,32]. In each patient, the power of these frequency bands was averaged within three-time epochs: (1) PRE-ONSET (PO), corresponding to 1 s of the countdown before the video’s onset; (2) STILL HANDS (SH), corresponding to 1 s of the static image at the beginning of the videos; (3) ACTION OBSERVATION (AO), corresponding to 2 s of action execution, starting at movement onset (see Figure 2).

### 2.4. Statistics

Statistical analyses were performed using Statistica (StatSoft Inc., Hamburg, Germany). A one-way analysis of variance (ANOVA) was performed to assess the effect of the factor TREATMENT (i.e., AOT, ICT, or RAT-FES) on the absolute improvement of motor performance from T0 to T1 in the FMA_UE, as well as on the normalised improvement in the same outcome measure. This metric is calculated by dividing the absolute improvement in FMA_UE from T0 to T1 by the maximum nominal improvement, that is, the difference between the maximum FMA_UE score (i.e., 66) and the FMA_UE at T0:T1 − T0/(66 − T0)

Instead of measuring the absolute change in motor performance during recovery, this metric indicates how close each patient has come to the ideal performance of a healthy individual. The ideal therapeutic approach indeed would restore the original motor function as it was before the stroke, independently from the level of impairment produced by the brain lesion. The normalised improvement allowed us to elucidate how much each treatment included in the experimental design approached this ideal outcome. For example, patient 18 and patient 28 underwent the same improvement as indexed by the FMA_UE score (Δ FMA_UA_T0−T1_ = 17). However, in patient 28 the FMA_UE at T1 approach that of a healthy individual (FMA_UA_T1_ = 59), while in patient 18 was considerably lower (FMA_UA_T1_ = 42). This means that in patient 28, the rehabilitative intervention (RAT-FES) was able to restore up to 71% of the lost motor function, while the approach used in patient 18 (ICT) only reduced upper limb disability by 41%. In the BBT, the maximum number of blocks included in the test is 150, which is largely above the number of blocks that healthy subjects can move in one minute (~70) [33]. Therefore, this assessment does not virtually have a nominal maximal improvement as the FMA_UE; we thus considered only the absolute variation from T0 to T1, which was entered in a one-way ANOVA to assess the effect of TREATMENT as performed for the FMA_UE.

Depending on the severity of the upper limb impairment, patients were further divided into two subgroups by adopting a cut-off of 19 in the FMA_UE (mild/moderate impairment: FMA_UE_T0_ ≥ 19; severe impairment: FMA_UE_T0_ < 19) [28]. Post-hoc independent-sample *t*-tests with Bonferroni’s correction for multiple comparisons were carried out to further assess any significant effect of TREATMENT.

A repeated-measures ANOVA was performed on the spectral power of the low-alpha, high-alpha, and beta frequency bands with factors WEEK (first, second, and third), SIDE (lesioned, and non-lesioned hemisphere), and EPOCH (PO, SH, AO). Significant main effects were evaluated by means of post-hoc multiple comparisons with Bonferroni’s correction. To further investigate the effect of SIDE during AO, independent-sample *t*-tests were conducted on each frequency band, comparing power suppression between the lesioned and the non-lesioned hemisphere both in mild/moderate and severe subgroups.

## 3. Results

### 3.1. Clinical Outcome

The majority of patients recruited in this study showed an overall improvement in motor function from T0 to T1 (Table 2). More specifically, in 36% of AOT patients, 42% of ICT patients, and 50% of RAT-FES patients (overall 42% of patients) this improvement was clinically relevant according to the minimal clinically important difference (MCID), that is, it was larger than nine points in the FMA_UE score [34]. The improvement in the clinical outcome did not yet show a significant effect of TREATMENT (*p* > 0.05, one-way ANOVA). This result may be biased by the uneven distribution of stroke severity among patients who underwent different rehabilitative interventions. Among those who received ICT and RAT-FES, 11/12 and 9/10 patients carried a mild/moderate impairment of the upper limb according to the cut-off score proposed by Woodbury and colleagues [28], respectively. In contrast, only one patient in each of these two treatment groups carries a severe impairment. On the other hand, 5/11 patients who received AOT showed a mild/moderate impairment, the others 6/11 being classified as severe. Because the clinical outcome reflects the integrity of the neural substrates after stroke [35,36], patients with milder motor impairment may benefit to a larger extent from rehabilitative approaches targeting residual neural plasticity. To put this to the test, we investigated the improvement of motor function from T0 to T1 after excluding patients with severe motor impairment from each group. This resulted in a significant effect of TREATMENT, both in the ΔFMA_UE (F_2,22_ = 4.192, *p* = 0.029, one-way ANOVA), and in the normalised improvement (F_2,22_ = 6.428, *p* = 0.006, one-way ANOVA) in the FMA_UE score between T0 and T1 (Table 3). Specifically, the improvement of mild/moderate patients who received AOT results significantly larger as opposed to both the ICT and the RAT-FES group (ΔFMA_UE: AOT vs. ICT *p* = 0.049, AOT vs. RAT-FES *p* = 0.041, Bonferroni-corrected post-hoc; normalised improvement: AOT vs. ICT *p* = 0.008, AOT vs. RAT-FES *p* = 0.015, Bonferroni-corrected post-hoc) (Figure 3, middle-right and upper-right panels respectively). No significant difference was detected between ICT and RAT-FES (ΔFMA_UE: *p* > 0.05, Bonferroni-corrected post-hoc; normalised improvement: *p* > 0.05, Bonferroni-corrected post-hoc) The improvement measured by the BBT was not significantly different between patients who received different treatments, neither when all patients were considered nor when the analysis was restricted to patients with mild/moderate motor impairments (Figure 3, lower panels). Nevertheless, in this latter case, the pattern of (non-significant) results resembled those seen for the FMA_UE (Figure 3, lower-right panel).

### 3.2. EEG Results

Not surprisingly [29,37], the spectral power of all the three frequency bands examined in the EEG signal recorded from the C3 and C4 electrodes showed a main effect of EPOCH (low-alpha: F_2,16_ = 22.535, *p* < 0.001; high-alpha: F_2,16_ = 44.16, *p* < 0.001; beta: F_2,16_ = 25.534, *p* < 0.001), being significantly lower in the AO epoch as opposed to SH and PO (all *p* < 0.001). In addition, the power of the low-alpha frequency band shows a significant effect of SIDE (F_1,8_ = 5.559, *p* = 0.046) and SIDE by EPOCH interaction (F_2,16_ = 9.862, *p* = 0.001). The suppression of low-alpha power during AO is greater in the non-lesioned vs. lesioned hemisphere (*p* < 0.001). No WEEK effect was detected in any frequency band. *T*-test showed that low-alpha suppression during AO was significantly weaker in the lesioned side than in the non-lesioned side in patients with severe impairment (t8 = 2.867, *p* = 0.020) (Figure 4), while the between-side difference was observed neither in patients with mild/moderate impairment, nor for other frequency bands, during the first week of treatment.

## 4. Discussion

Accumulating evidence suggests that AOT is an effective rehabilitative intervention in stroke survivors [19]. Here, we show that AOT proves as effective as ICT and RAT-FES in terms of motor recovery of the upper limb motor function and possibly even superior in patients with moderate stroke-related impairments. Indeed, patients who received AOT, ICT, and RAT-FES showed a similar improvement in motor function of the upper limb as indexed by FMA_UE. On the other hand, previous work demonstrated that rehabilitative interventions involving AOT are more effective than traditional approaches hinged on active motor execution [21,23,38,39]. Some methodological differences may yet help to frame this inconsistency of results. For example, some studies combined action observation with a larger extent of action execution (e.g., through the imitation of previously observed actions), resulting in a larger amount of physical training of the paretic upper limb [21,22,40]. By contrast, in the current study, patients were asked to observe an overall daily amount of 120 actions and to imitate only 12 among them. In turn, this approach can reduce fatigue (and possibly frustration) and, thus, promote patient compliance. At the same time, some previous work employed more prolonged AOT-based interventions (e.g., up to 8 weeks) [40,41] than the three-week approach used here.

An important (though still preliminary, due to the small sample size) result emerging from the present study is that the outcome of AOT is crucially influenced by the initial severity of motor impairment, being more effective in patients with mild/moderate disability of the upper limb as indexed by an FMA score ≥ 19 at T0 [28]. In this subgroup of patients, AOT proved more effective than ICT and RAT-FES. The 19 ± 2 cut-off in the FMA_UE scale was proposed by Woodbury and colleagues to stratify stroke survivors with different degrees of motor impairment but remains rarely used in clinical practice [28]. If this result is confirmed in studies with larger sample sizes, the future adoption of this cut-off may help to select those patients that are most likely to benefit from an AOT intervention. The limited representation of patients with severe motor impairment among those who received ICT and RAT-FES prevented us from ascertaining whether the degree of disability impacts also the outcome of these rehabilitative interventions. Further investigations are needed to address this point.

In contrast with FMA_UE, the performance on the BBT showed a similar improvement regardless of the rehabilitative strategy employed, also when only patients with mild/moderate impairments were considered. In this regard, an important difference between these two assessments resides in the degree of complexity of motor control that they explore. The FMA_UE examines basic elements of motor control (e.g., intransitive flexion/extension movements, pronation/supination, reflexes, grasping movements), while the BBT targets manual dexterity, being thus influenced by the execution of a complex motor chain involving reaching-grasping, lifting and transporting the blocks [33]. It is therefore reasonable that early improvements of the motor function occurring within three weeks from the beginning of rehabilitative interventions are better captured by the FMA_UE.

Additional differences between patients in the three treatment groups regard the type of stroke. All patients who received ICT and RAT-FES had an ischaemic stroke, whilst those who received AOT had either an ischaemic or haemorrhagic stroke. This disparity among groups has to be considered in the interpretation of our results, given that few studies indicated how haemorrhagic stroke recovered more rapidly in the first three months after a stroke [42], with a limited time window for acquiring independence in activities of daily living [43].

Even if the three groups were matched for time since the stroke, another potential factor that has to be taken into account in the interpretation of our results is the spontaneous recovery. It occurs mainly within the first weeks after a stroke and it can be responsible for the trajectories of the recovery obtained [44]. However, it is difficult to separate the observed time-dependent changes over time, discriminating how much they are due to biological processes or rehabilitation interventions and environments.

In addition to investigating the impact of AOT on upper limb motor recovery after stroke, we evaluated whether a well-established neurophysiological correlate of action observation (i.e., µ suppression) may predict the outcome of AOT in individual patients, thus helping to design more individualized rehabilitative interventions [45]. This may help to understand the reason for the outcome discrepancy between patients with different degrees of motor impairment. It is conceivable that this difference depends on the distinct amount of residual function of the MNS following the brain lesion. As the chance of motor recovery is largely constrained by the extent of brain damage, the outcome of AOT may be crucially influenced by patients’ residual capability to map the observed actions into their own motor repertoire. In line with previous results, µ suppression in central electrodes during action observation was especially vigorous in the low-alpha frequency band [31,46] and the non-lesioned hemisphere [47]. Most importantly, the hemispheric asymmetry in low-alpha power suppression was more pronounced in patients with severe motor impairment at T0. This suggests that in these patients the MNS may be more severely compromised, thereby limiting their ability to recruit movement-related cortical resources during action observation, and ultimately undermining the effectiveness of AOT. Previous work has shown some association between EEG signatures, stroke severity, and clinical outcome. For example, recent work demonstrated that the theta (3.5–7.5 Hz) power at rest predicts the clinical outcome in terms of independence in daily activities [48]. Similarly, functional connectivity assessed four weeks after stroke between EEG electrodes placed over ipsilesional motor-related brain areas is associated with FMA improvement at eight weeks [49]. Additional EEG-derived features that convey valuable prognostic information in stroke patients are the Brain Symmetry Index (i.e., a measure of spectral symmetry between the two hemispheres) and the Laterality Coefficient (i.e., the differential activation of the ipsilesional vs. contralesional hemisphere during motor imagery) [50,51]. Although the small sample size employed in this study limits the range of our speculations, our preliminary results suggest that the assessment of µ suppression asymmetry during action observation may represent a feasible approach to determine which patients might benefit most from AOT. Importantly, this approach requires a quite simple two-electrode EEG montage with minimal patient preparation. Moreover, in contrast with standardized assessments (such as the FMA_UE score), it is highly objective and does not require complex training for health care professionals involved in patients’ evaluation. 

This preliminary study has some limitations. Firstly, the non-randomized design prevents us to draw any conclusion on the effects of AOT in comparison with two well-established arm interventions. Moreover, the additional inclusion of a control group that received no specific intervention (wait and see) would help in defying the role of spontaneous recovery, even if this solution can be not feasible in an inpatient rehabilitation setting. However, the three rehabilitation strategies are barely comparable, requiring different types of upper limb movements and cognitive involvement during the treatment. However, the three groups were time-matched, meaning that the same amount (100 min) of diverse rehabilitation activities was added to their multidisciplinary conventional therapy. Secondly, the small number of patients in the AOT group with mild moderate impairments would prevent the generalisability of the statistically significant results. Nonetheless, further investigations, such as RCT confirmatory studies on bigger samples, should be conducted to reasonably compare AOT to other rehabilitative interventions to better understand its role in motor function recovery after stroke.

## 5. Conclusions

In this preliminary, non-randomized study, AOT showed similar arm motor recovery as previously investigated techniques ICT and RAT-FES, though seemingly more effective in stroke survivors with mild/moderate motor impairments. Moreover, EEG resulted in a valid (and low-cost) tool to assess the severity of stroke during the subacute phase, possibly also providing predictive information on the potential use of AOT at the individual-patient level. A practical suggestion from these results is that AOT might be effectively employed in patients with mild or moderate motor impairments, especially in the initial phase of the rehabilitative intervention when patients’ disabilities and low fatigue threshold may substantially limit the feasibility of approaches requiring active collaboration.

## Figures and Tables

**Figure 1 jcm-12-01327-f001:**
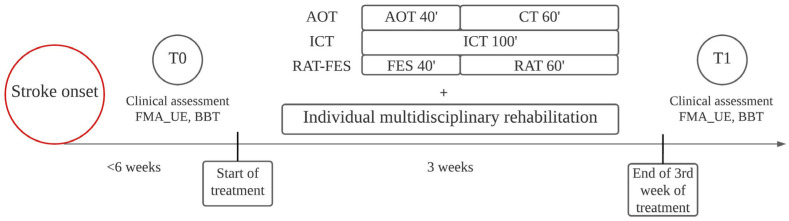
Timeline of the study. Patients were clinically evaluated and started treatments within six weeks of stroke onset. T0 marks the initial assessment before the beginning of treatments. T1 marks the clinical assessment after three weeks of treatment. As well as AOT, ICT, and RAT-FES, each patient received individual rehabilitation based on personal needs.

**Figure 2 jcm-12-01327-f002:**
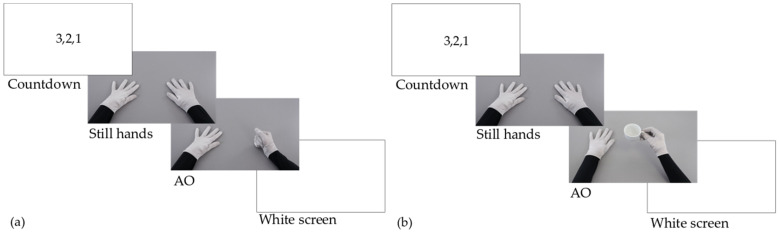
AOT treatment. Each video was preceded by a 2-s countdown and started with a 1-s image with still hands relaxed on a table. (**a**) Example intransitive action during AO: Hitting the table with a fist. (**b**) Example transitive action during AO: Grasping and lifting a teacup.

**Figure 3 jcm-12-01327-f003:**
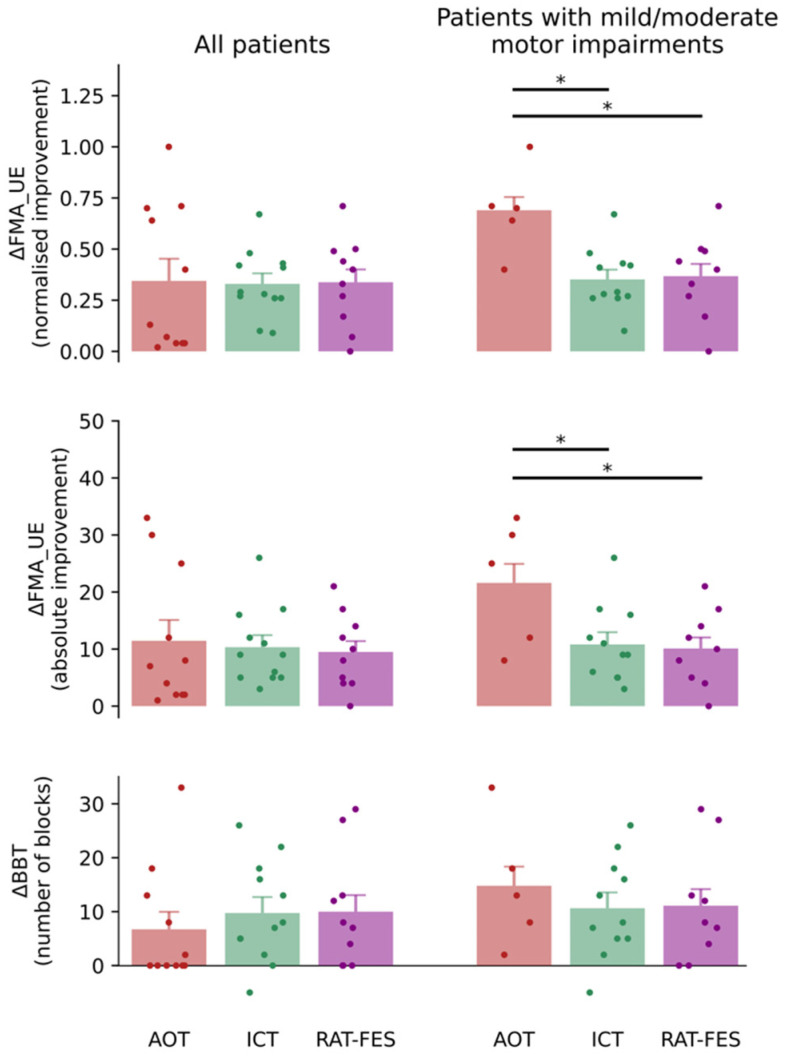
Normalised improvement in the FMA_UE (upper panels), absolute improvement in the FMA_UE (middle panels), and variation in the BBT from T0 to T1 (lower panels) for patients who received the three rehabilitative interventions (AOT, ICT, RAT-FES). The panels on the left and right depict the pooled data of all patients and the data of patients with mild/moderate motor impairments, respectively. The three rehabilitative interventions yielded similar motor function improvement when all patients’ pooled data were considered. In contrast, when the analysis was restricted to patients with mild/moderate motor impairments, a significant difference was found in the normalized improvement and in the absolute improvement in the FMA_UE between those treated with AOT and those who received ICT and RAT-FES. The BBT variation exhibits a similar pattern of result, although not reaching significance. Vertical bars indicate the standard error of the mean (SEM). The coloured dots indicate data from individual patients. Horizontal bars and asterisks denote significant improvement from T0 to T1 (i.e., *p* < 0.05).

**Figure 4 jcm-12-01327-f004:**
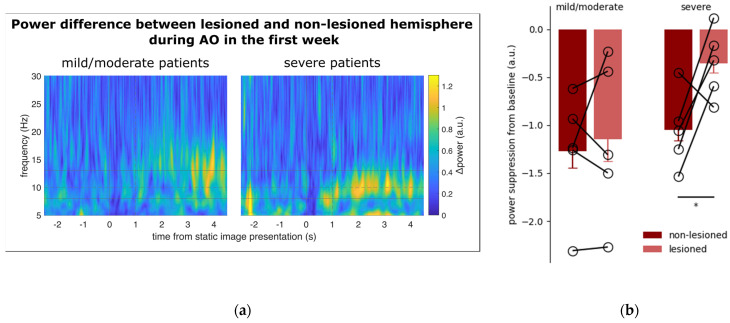
(**a**) Time course of the spectral power difference between the lesioned and non-lesioned hemisphere in patients with mild/moderate (left) and severe motor impairments. The data are aligned at zero with the onset of the still hands (see Methods). The vertical dashed line at +1 s denotes movement onset. The continuous horizontal lines illustrate the frequency band from low-alpha to high-alpha. The dashed horizontal line marks the boundary between low- and high-alpha (i.e., 10 Hz). In patients with severe motor impairment, the power difference between the lesioned and non-lesioned hemispheres is markedly enhanced during action observation, especially in the low-alpha frequency band. (**b**) Power suppression from baseline during action observation in the C3/C4 electrodes positioned over the non-lesioned and lesioned hemispheres of patients with mild/moderate (left) and severe (right) motor impairments. In severe patients, suppression showed a significant asymmetry (i.e., *p* < 0.05), marked with the asterisk, being more pronounced in the non-lesioned hemisphere. No significant differences were detected in mild/moderate patients. Vertical bars represent SEM. The empty circles connected by sticks indicate data from individual patients.

**Table 1 jcm-12-01327-t001:** Demographic and clinical data.

#Patient	Age	Sex	Plegic Side	Days from Stroke Onset	Type of Stroke	Severity *	Group
1	49	M	Left	31	Haemorrhagic	mild/moderate	AOT
2	59	M	Left	32	Haemorrhagic	mild/moderate	AOT
3	61	F	Left	14	Haemorrhagic	mild/moderate	AOT
4	77	M	Right	26	Ischaemic	severe	AOT
5	57	F	Left	27	Haemorrhagic	mild/moderate	AOT
6	79	M	Left	36	Ischaemic	severe	AOT
7	55	M	Left	23	Ischaemic	severe	AOT
8	50	F	Left	33	Ischaemic	severe	AOT
9	48	M	Left	25	Ischaemic	severe	AOT
10	60	M	Right	9	Haemorrhagic	mild/moderate	AOT
11	49	M	Left	41	Haemorrhagic	severe	AOT
12	66	F	Right	41	Ischaemic	mild/moderate	ICT
13	58	M	Right	37	Ischaemic	mild/moderate	ICT
14	73	F	Left	10	Ischaemic	mild/moderate	ICT
15	73	M	Left	18	Ischaemic	mild/moderate	ICT
16	77	M	Right	11	Ischaemic	severe	ICT
17	74	M	Right	18	Ischaemic	mild/moderate	ICT
18	45	M	Left	12	Ischaemic	mild/moderate	ICT
19	67	F	Left	24	Ischaemic	mild/moderate	ICT
20	69	F	Left	28	Ischaemic	mild/moderate	ICT
21	59	M	Left	39	Ischaemic	mild/moderate	ICT
22	71	F	Right	22	Ischaemic	mild/moderate	ICT
23	23	M	Right	24	Ischaemic	mild/moderate	ICT
24	68	F	Left	23	Ischaemic	mild/moderate	RAT-FES
25	68	M	Right	21	Ischaemic	mild/moderate	RAT-FES
26	39	F	Left	21	Ischaemic	severe	RAT-FES
27	79	M	Left	12	Ischaemic	mild/moderate	RAT-FES
28	43	M	Left	25	Ischaemic	mild/moderate	RAT-FES
29	55	M	Left	17	Ischaemic	mild/moderate	RAT-FES
30	59	F	Right	14	Ischaemic	mild/moderate	RAT-FES
31	71	F	Left	26	Ischaemic	mild/moderate	RAT-FES
32	62	M	Right	39	Ischaemic	mild/moderate	RAT-FES
33	69	F	Left	28	Ischaemic	mild/moderate	RAT-FES

* According to [28].

**Table 2 jcm-12-01327-t002:** FMA_UE and BBT at T0 and T1 and variation (ΔFMA_UE and normalised improvement for the FMA_UE) from T0 to T1 in each patient.

#Patient	FMA_UE T0	FMA_UE T1	ΔFMA_UE	Normalised Improvement	BBT T0	BBT T1	Δ BBT	TREATMENT
1	19	49	30	0.64	0	18	18	AOT
2	46	54	8	0.4	17	19	2	AOT
3	31	56	25	0.71	2	35	33	AOT
4	9	10	1	0.02	0	0	0	AOT
5	54	66	12	1	47	55	8	AOT
6	11	15	4	0.07	0	0	0	AOT
7	13	15	2	0.04	0	0	0	AOT
8	12	14	2	0.04	0	0	0	AOT
9	14	21	7	0.13	0	0	0	AOT
10	19	52	33	0.7	0	13	13	AOT
11	13	15	2	0.04	0	0	0	AOT
12	27	53	26	0.67	11	29	18	ICT
13	43	49	6	0.26	33	40	7	ICT
14	28	44	16	0.42	6	32	26	ICT
15	37	40	3	0.1	9	25	16	ICT
16	13	18	5	0.09	0	0	0	ICT
17	41	53	12	0.48	30	25	−5	ICT
18	25	42	17	0.41	0	13	13	ICT
19	47	52	5	0.26	16	38	22	ICT
20	49	54	5	0.29	47	52	5	ICT
21	33	42	9	0.27	0	5	5	ICT
22	26	37	11	0.28	0	2	2	ICT
23	45	54	9	0.43	22	30	8	ICT
24	37	37	0	0	3	7	4	RAT-FES
25	51	56	5	0.33	28	41	13	RAT-FES
26	12	16	4	0.07	0	0	0	RAT-FES
27	23	44	21	0.49	13	25	12	RAT-FES
28	42	59	17	0.71	20	47	27	RAT-FES
29	41	51	10	0.4	16	23	7	RAT-FES
30	38	52	14	0.5	12	41	29	RAT-FES
31	43	47	4	0.17	25	33	8	RAT-FES
32	48	56	8	0.44	30	30	0	RAT-FES
33	21	33	12	0.27	0	0	0	RAT-FES

**Table 3 jcm-12-01327-t003:** Mean ± SD of clinical data in the three groups of treatments.

	All Patients (Mean ± SD)			Patients with Mild/Moderate Impairment (Mean ± SD)		
	AOT	ICT	RAT-FES	AOT	ICT	RAT-FES
FMA_UE T0	21.91 ± 15.21	34.5 ± 10,97	35.6 ± 12.7	33.8 ± 15.83	36.45 ± 9.05	38.22 ± 10.21
FMA_UE T1	33.36 ± 21.63	44.83 ± 10.41	45.1 ± 13.24	55.4 ± 6.47	47.27 ± 6.37	48.33 ± 8.92
Δ FΜA_UΕ	11.45 ± 12.07	10.33 ± 6.65	9.5 ± 6.57	21.6 ± 11.06 *	10.82 ± 6.75	10.11 ± 6.66
Normalised Improvement FMA_UE	0.34 ± 0.36	0.33 ± 0.16	0.34 ± 0.22	0.69 ± 0.21 *	0.35 ± 0.15	0.37 ± 0.21
BBT T0	6 ± 14.51	14.5 ± 15.53	14.70 ± 11.19	13.2 ± 20.19	15.82 ± 15.57	16.33 ± 10.52
BBT T1	18.23 ± 12.73	24.25 ± 16.25	24.70 ± 17.20	28 ± 17.2	26.45 ± 15.04	27.44 ± 15.75
Δ ΒΒΤ	6.73 ± 10.73	9.75 ± 9.34	10 ± 10.60	14.8 ± 11.78	10.64 ± 9.25	11.11 ± 10.61

Asterisks denote significant improvement from T0 to T1 (i.e., *p* < 0.05).

## Data Availability

The data that support the findings of this study are available from the corresponding author upon reasonable request.

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
