# Peer review of "Action Observation Therapy for Arm Recovery after Stroke: A Preliminary Investigation on a Novel Protocol with EEG Monitoring"

_jcm, 2023, doi:10.3390/jcm12041327_

Round 1

Reviewer 1 Report

In this preliminary study by Boni and colleagues they test a fairly novel treatment, action observation therapy (AOT), in subacute stroke patients and compare the efficacy in terms of motor recovery with intensive conventional therapy and robot assisted therapy combined with FES.  They go on to propose the mechanism of action for this therapy using mirror neurons and assess for mirror neuron activation with EEG.  While the treatment concept was novel and intriguing, there were major flaws in the study design that likely biased the results, namely the use of the normalized improvement metric as the primary outcome measure.  In addition, the authors need to at least acknowledge some major limitations of the study including the lack of an observation only control group and the more rapid recovery in hemorrhage patients during the subacute phase reported in the literature.

Major:

1.  The “normalized improvement” metric is not a valid primary outcome measure, and likely biased the results of this study.  For example, patient #1 went from FMA_UA T0 of 19 to FMA_UA T1 of 49 for a delta FMA_UA of 30.  Meanwhile patient #5 went from FMA_UA T0 of 54 to FMA_UA T1 of 66 for a delta FMA_UA of 12.  A 30 point change in FMA_UA for a patient who starts off with a greater level of motor impairment is much more clinically meaningful than a 12 point change in someone with mild impairment.  Nonetheless, patient #1 had a much worse normalized impairment score (0.64) than patient #5 (1.0).  While the raw delta FMA_UA is not perfect, it is much better than the normalized improvement score.  In addition, by the nature of the normalized improvement score, the more impaired patients are automatically at a disadvantage because their recovery will be counted less.

2.  There is no control group receiving standard care for comparison.  At this timepoint post-stroke there is a great deal of spontaneous biological recovery and many of these patients would improve considerably even if no intervention were delivered.  Do the investigators have any historical controls to add for comparison?

3.  The authors note that most of the AOT patients had hemorrhages whereas patients in the other 2 cohorts all had ischemic stroke.  They go on to suggest in the discussion that patients with hemorrhagic stroke recover with similar speed and magnitude to those with ischemic stroke, but this is not true.  In fact, the paper they cite by Perrson et al. shows that hemorrhagic stroke patients recover more quickly than ischemic stroke patients from day 10 to 3 months (figure 3).  There is another paper by Schepers et al. (see below) showing similar findings.  If hemorrhage patients recover more quickly in the subacute phase, this alone could explain their findings.

Schepers VP et al. Functional recovery differs between ischaemic and haemorrhagic stroke patients. Journal of rehabilitation medicine. 2008;40:487-489.

Minor:

1. There needs to be a table either in the main body of the journal or in supplementary material showing the mean T0, T1, and delta FMA_UE and BBT scores for all AOT, ICT, and RAT-FES as well as for the subgroups with only mild impairment.  While some of this is shown in figure 3 graphically, we do not know where the patients started out in terms of level of impairment, making it difficult to know whether the groups were comparable at baseline.

2. In the results it is unclear what, “all groups demonstrated a clinically significant arm function improvement” means exactly.  All patients did not achieve the MCID, so are the authors saying the mean change was greater than the MCID for each study cohort?

Reviewer 2 Report

“Action observation therapy for arm recovery after stroke: A preliminary investigation on a novel protocol with EEG monitoring “.

    This study addresses an important issue in Stroke rehabilitation. It has been suggested that motor recovery after stroke results as a consequence of neuroplasticity. Action observation has been suggested as a promising therapeutic strategy for upper limb motor rehabilitation after stroke. It is considered a multisensory approach, involving both somatosensory and cognitive rehabilitation through activation of the mirror neural system (MNS) of the brain (Johansson 2011). A systematic review published in 2018 investigated the effects of action observation on motor function and upper limb motor performance and cortical activation in people with stroke, found evidence that AO is beneficial in improving upper limb motor function and dependence in activities of daily living (ADL) in people with stroke, when compared with any control group. However, the quality of the evidence was low (Borges 2018). Therefore, future research on this treatment strategy is needed.   The current study objective was to compare movement therapy and robot-assisted therapy with action observation therapy (AOT). The manuscript is well reasoned and well written. However, there are minor limitations, which should be addressed for the manuscript to be considered for publication. Considering the information that has to be reported in a controlled trial, I do recommend that the authors try, if possible, to address the following issues and re-submit the manuscript. Here are my comments:

  Although the authors report that this is a preliminary study, it is not clear to the reader that this is a non-randomized trial. Participants in the two comparison groups were previously randomly allocated. However, participants in the AOT group were recruited later. This fact must be considered as an important limitation of the study. It was not clear if the examiners were blinded after assignment to interventions. In addition, since the sample size was not determined, the term “efficacy” should be used with caution. The following reference should be considered in the introduction and discussion sections:

Borges LR, Fernandes AB, Melo LP, Guerra RO, Campos TF. Action observation for upper limb rehabilitation after stroke. Cochrane Database Syst Rev. 2018 Oct 31;10(10):CD011887. doi: 10.1002/14651858.CD011887.pub2. Update in: Cochrane Database Syst Rev. 2022 Aug 5;8:CD011887. PMID: 30380586; PMCID: PMC6517007.

Round 2

Reviewer 1 Report

1. Need to add a delta FMA-UE to table 2 between the FMA_UE T1 and the Normalised Improvement columns

2. Add another graph for delta FMA-UE to figure 3

3. Sentence lines 259 to 262 – Unclear whether this is in regard to delta FMA-UE or the normalised score, or perhaps the results are the same for both?  Seems the effect for TREATMENT should be different for delta FMA-UE and the normalised score (lines 256-258), but perhaps they end up the same because they are mathematically coupled.

4. Line 323-325 of the discussion – the concept of proportional recovery where most patients recover around 70% of the function lost has now been thoroughly debunked in several papers due to issues with mathematical coupling.  This 2017 paper by Stinear and colleagues is now outdated as a result.  The authors should try to make their argument without invoking proportional recovery.  (no need to quote the paper below in the manuscript, just including here for the author’s reference)

pe T, Bzdok D, Guggisberg AG, Hawe RL, Dukelow SP, Rehme AK, Fink GR, Grefkes C, Bowman H. Bringing proportional recovery into proportion: Bayesian modelling of post-stroke motor impairment. Brain : a journal of neurology. 2020;143:2189-2206 

5. The authors need to add to the limitations paragraph (lines 370-380) that the study would have benefited from an observation only control group to compare to the results of spontaneous recovery.  This paragraph should also acknowledge the small number of patients (5) in the AOT group with mild/moderate impairment which led to the statistically significant results.
